# Measurement and Influencing Factors of Low Carbon Urban Land Use Efficiency—Based on Non-Radial Directional Distance Function

**Han Chen** [1] , **Chunyu Meng** [2] **and Qilin Cao** [2],*

[1] Financial Sector, Beijing Dentons Law Offices, LLP (Chengdu), Chengdu 610021, China; 2017225029472@stu.scu.edu.cn
[2] Business School, Sichuan University, Chengdu 610064, China; 2021225020115@stu.scu.edu.cn
* Correspondence: qlcao@scu.edu.cn

**Abstract:** The development and use of urban land is a major source of carbon emissions. How to reduce carbon emissions in the process of urban land use without harming the economy has become an extremely important issue. This paper integrating carbon emissions into the urban land use efficiency evaluation index system, measures low-carbon urban land use efficiency using a non-radial directional distance function and analyses its spatial and temporal evolution and its influencing factors using a combination of a kernel density estimation method and a Tobit model. The study found that: (1) China's low-carbon urban land use efficiency shows a fluctuating development and tends to converge; (2) there is much room for reducing land input and carbon emissions in China, and in 2016 alone, land input and carbon emissions in the sample could be reduced by 10.38% and 5.31%, respectively; (3) at the national level, land finance, economic level and population density have a positive impact on low-carbon urban land use efficiency, while the traffic level has negative effects, and these effects show regional heterogeneity. Accordingly, the paper proposes corresponding policy recommendations.

**Keywords:** low carbon; low-carbon urban land use efficiency; non-radial directional distance function





## 1. Introduction

In recent years, urban land use issues have received increasing attention in the face of a global wave of urbanization [1]. On the one hand, rapid urbanization is accompanied by a rapid expansion of urban land, which usually comes at the expense of large amounts of agricultural and environmental land, posing a threat to food security and the ecological environment [2–4]. On the other hand, urban land is the spatial carrier of urbanization, carrying the development of economic, social, political and cultural activities in the city, and its efficiency of use directly affects the social and economic development of the region and the construction of the human living environment [5–7]. Therefore, governments and scholars are actively exploring and practicing ways to improve the efficiency of urban land use, especially the Chinese government.

Since the reform and opening up, China's urbanization level has risen from 17.92% in 1978 to 63.89% in 2020 [8]. However, this satisfactory result has been accompanied by an egregious waste of land resources and a dramatic increase in pollutant emissions (e.g., $CO_2$ emissions) [9]. Through 2017, China's urban construction land had reached 55,155.5 square km [10]. In contrast, the amount of arable land per capita has been declining, from 0.16 $hm^2$ to 0.09 $hm^2$ from 1961 to 2015, which is consistently lower than the world's per capita arable land (0.37 $hm^2$ in 1961, 0.19 $hm^2$ in 2015) [11]. The central government has controlled the rapid growth of built-up areas through various administrative methods, for example, arable land warning lines have been set and land quota systems have been implemented [12]. However, due to the demands of economic development

and urbanization, the urban land area continues to grow at a high rate, and the problems of disordered urban land expansion and inefficient urban land use are still serious. A reasonable assessment of urban land use efficiency has become the key to improving urban land use efficiency and reducing the negative impacts arising from the urban land use process [13].

In addition, $CO_2$ emissions are one of the main issues facing the Chinese government. In 2007, China surpassed the United States to be the largest emitter of carbon dioxide in the world [14]. Excessive $CO_2$ emissions contribute to climate warming and thus increase the occurrence of extreme weather, posing a threat to the ecological environment and human survival [15]. Urban land expansion is considered to be the main source of $CO_2$ emissions [16,17]. In 2017, China's urban carbon emissions reached 70% of the national carbon emissions [18]. To reduce carbon emissions, the Chinese government has taken several initiatives, such as implementing a low-carbon city pilot policy, building a carbon emissions trading market and committing to reduce carbon emissions intensity by 60–65% in 2030 relative to 2005 [19–21]. Therefore, incorporating carbon emissions into the evaluation system of urban land use efficiency not only enables low-carbon use of urban land, but also facilitates the development of a low-carbon urban economy, thus helping China to reach its set strategic goals.

Based on this background, this study constructs an evaluation system for the Low Carbon Urban Land Use Efficiency (LCULUE) by taking carbon emissions as a non-desired output, measures the LCULUE in each city using a non-radial directional distance function based on the statistical data of 204 cities in China from 2005 to 2016 and explores its dynamic evolution pattern and influencing factors using the kernel density estimation method and the Tobit model. The possible contributions of this study include: First, from the perspective of low-carbon urban land use, an evaluation system for low-carbon land use efficiency in Chinese cities is constructed, which integrates economic development, land conservation and low-carbon development, providing a new perspective for examining urban land use efficiency in China and providing a useful supplement to existing studies. Second, differing from the conventional measurement of all-factor urban land use efficiency, this study uses a non-radial distance function to measure urban land use efficiency, which separates the inefficient effects of other input factors and can better highlight the characteristics of urban land inputs, enriching the measurement method of urban land use efficiency. Third, the dynamic evolution and influencing factors of urban land use efficiency are analyzed. This has policy guidance implications for Chinese cities to promote efficient urban land use and achieve low-carbon economic development.

The framework of the remainder of the study is as follows: Section 2 provides a brief review of the relevant literature; Section 3 introduces the measures of LCULUE and related models and provides a brief description of the data; Section 4 reviews and briefly analyses the empirical results; and Section 5 draws conclusions and makes corresponding policy recommendations.

## 2. Literature Review

At present, a great number of studies have been conducted on urban land use efficiency in academia. From the analysis of urban land use efficiency to the measurement method, there are still many theories, but it is hard to say which one is better. In terms of the analysis of urban land use efficiency, some scholars believe that urban land use efficiency is a reflection of the comprehensive utility of land use, mainly by constructing an index evaluation system to measure urban land use efficiency [22,23]. The research perspective is more focused on input–output, and urban land use efficiency is considered to be the ratio between a variety of input combinations including urban land elements and output combinations [24,25], and the selection of corresponding indicators varies slightly, depending on the focus of the study. The simplest form of this is to use urban land as the sole input indicator and economic efficiency as the output indicator. The value of the increase in industry per unit of construction land area is used to express urban land use

efficiency [26,27]. This approach ignored the effect of other factors of production, and subsequent studies gradually included other factors of production such as capital and labor among the input factors [28,29]. With the emphasis on the concept of green development, numerous studies have also included other undesired outputs such as pollutants produced during the production process into the output mix to reflect the ecological impact of the land use process. It is emphasized that the negative impact on the ecological environment should be reduced while safeguarding economic development and social welfare in the land use process [10,30]. At the same time, research results show that models that do not take into account non-desired outputs tend to overestimate the level of urban land use efficiency compared to models that consider non-desired outputs [1,25].

In terms of methods for measuring urban land use efficiency, there are two main types of methods, parametric and non-parametric. The parametric method is the stochastic frontier production function (SFA) model, which introduces a random error term into the production function and estimates the efficiency value by regressing the production function with the advantage that the random factor in the compound error can be separated out. However, its accuracy depends on the accurate choice of the distribution of the error term and the correct setting of the form of the production function, which makes it more difficult to apply [31,32]. Non-parametric methods are various data envelopment analysis (DEA) models, specifically those that use a linear combination of existing production units to form a production frontier and then measure efficiency values by comparing the relative distance of each production unit to the production frontier [33].The DEA model has become the main method of efficiency assessment and has been perfected continuously due to its advantages of not requiring a specific functional form and not requiring access to price information of input and output factors [34]. In addition, research dimensions have become increasingly diverse, ranging from typical cities [35,36] to provincial cities [29,37], to urban agglomerations [38,39], national megacities [40] and other types of spatial scales.

A preliminary review of the above urban land use efficiency studies shows that the current research methods and perspectives are diversified, establishing a good theoretical and practical foundation for further research on urban land use efficiency, but there is still much potential for progress. First, from the perspective of research, there are few studies based on a low-carbon perspective, which include carbon dioxide generated in the process of urban land use as a non-desired output in the evaluation system; second, most existing studies use SFA and DEA models to measure urban land use efficiency, but indeed, whether it is the SFA model or the commonly used DEA model, the measured efficiency is all-factor efficiency, which cannot be distinguished from single factors such as land, labor and capital. If the efficiencies of other factors of production cannot be separated out, the characteristics of land cannot be emphasized; finally, as far as the dimension of research is concerned, most studies focus on local areas, and few studies cover the whole country, which cannot comprehensively reflect the current situation of land use in China. In this paper, from a low-carbon perspective, carbon dioxide is included in the evaluation index system of LCULUE, and a non-radial distance function is used to measure the nationwide LCULUE to separate out the influence of other factors of production and analyze the temporal trends, spatial distribution characteristics and influencing factors of urban land use efficiency to improve the urban land use efficiency of each city. The aim is to provide a scientific basis for decision making to improve urban land use efficiency.

## 3. Methodology and Data

### 3.1. Measurement Model for LCULUE

#### 3.1.1. Non-Radial Directional Distance Functions

Assuming that there are N cities as decision-making units (DUMs) with a total of T observation periods, each DUM produces in each period with input $x \in R^o_+$ factors to obtain a desired output $y \in R^p_+$ accompanied by a non-desired output $z \in R^q_+$, where the superscripts *o*, *p* and *q* denote the type of input, desired output and non-desired output,

respectively, $R_+$ is the set of positive real numbers. The set of production possibilities $P$ can be expressed as Equation (1).

$$P = \{(x, y; z) : x \ can \ produce \ y \ with \ z\}, \tag{1}$$

This production possibility set is a tight set; that is, limited inputs can only produce a limited output. At the same time, the production possibility set $P$ needs to satisfy the following two theorems [41,42] in addition to the basic axioms of the traditional production function [43].

(1) The desired output and the non-desired output combined set needs to satisfy weak disposability. That is, if $(x, y; z) \in P$ and $0 \leq \theta \leq 1$, then $(x, \theta y; \theta z) \in P$. This theorem shows that if non-desired outputs are to be reduced, desired outputs must be reduced at the same time. This shows that there is a cost to the reduction of pollutants.

(2) Zero intersection of desired and non-desired outputs. That is, if $(x, y; z) \in P$ and $z = 0$, then $y = 0$. This theorem shows that emissions of non-desired outputs from the production process are unavoidable, which means that desired and non-desired outputs are produced simultaneously and that production can only be stopped if no emissions of non-desired outputs are required.

On the basis of satisfying the above theorems, the production possibility set is constructed in this study. To make LCULUE comparable between years, this study uses all sample points within the study period to construct the production frontier surface [44]. Classical economic growth theories usually consider capital and labor as the basic elements of economic development, which ignores the function of the land element or considers it as part of capital by default. In fact, land is different from general capital, and it plays an important role in the economic development of China. At the same time, the $CO_2$ emissions that accompanied the development process of urban land use are rarely included in the evaluation system of urban land use efficiency. With low-carbon economic development becoming a strategic goal in China, it has become extremely important to balance economic development and $CO_2$ emissions in the process of urban land use development. Where land ($L$), capital ($K$) and labor ($LA$) are selected as input factors, economic output ($Y$) is used as a measure of desired output, and carbon dioxide is used as non-desired output ($C$), the production possibility set can be expressed as Equation (2).

$$P = \left\{ \begin{array}{l} (L, K, LA, Y; C) : \sum_{t=1}^{T} \sum_{i=1}^{N} \lambda_{it} L_{it} \leqslant L, \sum_{t=1}^{T} \sum_{i=1}^{N} \lambda_{it} K_{it} \leqslant K \\ \sum_{t=1}^{T} \sum_{i=1}^{N} \lambda_{it} LA_{it} \leqslant LA, \sum_{t=1}^{T} \sum_{i=1}^{N} \lambda_{it} Y_{it} \geqslant Y, \\ \sum_{t=1}^{T} \sum_{i=1}^{N} \lambda_{it} C_{it} = C, \lambda_{it} \geqslant 0 \end{array} \right\}, \tag{2}$$

where $t$ indicates the year, in this case 2005 is noted as 1, and so on; $T$ is the upper limit of the year, 12; $i$ indicates the ith city; $N$ is the upper limit of the number of cities, up to 204; and $\lambda$ indicates the weight of the observation.

A distance function is now defined for measuring land use efficiency in each DUM city. Traditional DEA uses the Shephard Distance Function (SDF), which assumes that all outputs will increase or decrease in same ratio [45], which does not satisfy our need to ensure that the desired output increases while the undesired output is reduced. (DDF) proposed by Chung et al. remedies this deficiency by allowing the desired output to increase while the undesired output decreases in the same ratio as far as the technically feasible set allows [41]. However, this may suffer from a "relaxation bias" [46]. Zhou et al. loosened this restriction by proposing a non-radial directional distance function (NDDF) that allows the two types of outputs to increase or decrease at different rates while also avoiding the possible relaxation bias problem of the DDF [47]. Drawing on Zhou et al., the NDDF is constructed in this study as shown in Equation (3) [47].

$$\overrightarrow{NDDF}(L, K, LA, Y, C; g) = \sup\{w^T \beta : \{((L, K, LA, Y, C) + g \cdot diag(\beta)) \epsilon P\}, \tag{3}$$

where $g = (g_L, g_K, g_{LA}, g_Y, g_C)$ is a direction vector to indicate the direction of the desired output increase and the non-desired output and input factor decrease. Here, $w = (w_L, w_K, w_{LA}, w_Y, w_C)^T$ is a weight vector to indicate the relative importance of each input and output factor, which can be set in advance according to the needs of the study and has good flexibility [48]. In addition, $\beta = (\beta_L, \beta_K, \beta_{LA}, \beta_Y, \beta_C)^T \geq 0$ is the slack vector, indicating the proportion of each input and output factor that can be increased or decreased. Finally, diag($\beta$) indicates the diagonalization of the vector $\beta$.

Because of the substitutability between input factors, the net efficiency value of land inputs cannot be measured without separating out the compressible proportions of capital and labor. In other words, we need to measure the maximum compressible ratio of land input and undesired output and the maximum expandable ratio of desired output when capital and labor are constant. For this reason, the weight variable $w = (1/3, 0, 0, 1/3, 1/3)^T$ is chosen in this study. This is due to the fact that in the absence of other a priori information, it may be more reasonable to treat the various input–output factors equally in the construction of the total factor indicator [49], so that the proportions of inputs, desired output and non-desired output are all 1/3. To separate the effects of capital and labor, the corresponding direction vector is $g = (-g_L, 0, 0, g_Y, -g_C)$.

Based on the above assumptions, the following DEA model is constructed in this paper, as shown in Equation (4).

$$
\begin{aligned}
\overrightarrow{NDDF}(L, K, LA, Y, C; g) = &\; max\left(\tfrac{1}{3}\beta_L + \tfrac{1}{3}\beta_Y + \tfrac{1}{3}\beta_C\right) \\
s.t. &\sum_{i=1}^{N}\sum_{t=1}^{T} \lambda_{i,t} L_{i,t} \leqslant L - \beta_L g_L \\
&\sum_{i=1}^{N}\sum_{t=1}^{T} \lambda_{i,t} K_{i,t} \leqslant K \\
&\sum_{i=1}^{N}\sum_{t=1}^{T} \lambda_{i,t} LA_{i,t} \leqslant LA \\
&\sum_{i=1}^{N}\sum_{t=1}^{T} \lambda_{i,t} Y_{i,t} \geqslant Y + \beta_Y g_Y \\
&\sum_{i=1}^{N}\sum_{t=1}^{T} \lambda_{i,t} C_{i,t} = C - \beta_C g_C \\
&\lambda_{i,t} \geqslant 0, i = 1, 2, \cdots, N \\
&t = 1, 2, \cdots, T, \beta_L, \beta_Y, \beta_C \geqslant 0,
\end{aligned}
\tag{4}
$$

The meaning of each symbol in Equation (4) is the same as Equations (2) and (3) and will not be repeated here. Based on Equation (4), the optimal solution is $\beta^* = (\beta_L^*, \beta_Y^*, \beta_C^*)^T$. If the city, $i$, achieves optimal production in year $t$, the target values of land input, desired output and undesired output are: $L_{it} - \beta_{L,it}^* \times L_{it}$, $Y_{it} + \beta_{Y,it}^* \times Y_{it}$ and $C_{it} - \beta_{C,it}^* \times C_{it}$. Obviously, when $\beta_{j,it}^* = 0$ $(j = L, Y, C)$, it indicates that urban $i$ has reached the optimum in terms of $j$ input (output) in year $t$. Based on the optimal solution, this study calculated the LCULUE, which is calculated as in Equation (5).

$$
\text{LCULUE}_{it} = \frac{1}{2}\left[\frac{\frac{L_{it} - \beta_{L,it}^* \times L_{it}}{Y_{it} + \beta_{Y,it}^* \times Y_{it}}}{\frac{L_{it}}{Y_{it}}} + \frac{\frac{C_{it} - \beta_{C,it}^* \times C_{it}}{Y_{it} + \beta_{Y,it}^* \times Y_{it}}}{\frac{C_{it}}{Y_{it}}}\right] = \frac{1 - \frac{1}{2}\left(\beta_{L,it}^* + \beta_{C,it}^*\right)}{1 + \beta_{Y,it}^*},
\tag{5}
$$

It can be seen that the LCULUE takes a value between 0 and 1. The higher the value, the more efficient the city's urban land use is, and when LCULUE = 1, the city is at the border of the production frontier, which is the highest level of efficiency.

### 3.1.2. Selection of Indicators and Description of Variables

As mentioned in the previous section, three input factors are selected in this study: land ($L$), capital ($K$) and labor ($LA$). The outputs are divided into desired and non-desired outputs, where the desired output is economic output, and the non-desired output is $CO_2$. The indicators are explained as follows and shown in Table 1.

**Table 1.** Input and output indicators of LCULUE.

| Variable Type | | Index |
|---|---|---|
| Input | Land | Built-up area |
| | Capital | Capital stock in urban areas |
| | Labor | Number of people employed in secondary and tertiary industries |
| Output | Economic | Gross regional product of secondary and tertiary industries |
| | Undesired | Carbon dioxide emissions in urban areas |

Land input: The area of built-up areas in municipal districts was chosen to measure land input. Most studies use the area of urban construction land to measure land input. In fact, the area of urban construction land is the area of land available for use in planning, while the area of built-up area is the area of actual construction land, which better reflects the real situation of land input [50].

Capital input: The capital stock of the municipal area was selected to measure capital input. As there is no data on urban capital stock in various statistical sources, this study uses the perpetual inventory method to calculate the capital stock of each share, as described in Tang et al. [25].

Labor input: The number of persons employed in secondary and tertiary industries in urban units in the municipal area is selected to measure labor input. In this study, it is assumed that there is no primary industry in built-up areas, rather than no secondary or tertiary industry in non-built-up areas. This is more reasonable, and the measurement error caused is acceptable, considering that secondary and tertiary industries are mainly concentrated in cities [29].

Economic output: The gross regional product of the secondary and tertiary industries in the municipal area was chosen to measure economic output. The reasons for this approach are the same as for the labor force.

Non-desired output: The city-level $CO_2$ emissions were chosen to measure the non-desired output. Current calculations of regional $CO_2$ emissions focus on the provincial level and the city level [51,52], while the scope of this study is limited to municipalities. Using city-level $CO_2$ emissions data would overestimate $CO_2$ emissions. The PSO-BP algorithm unifies the scales of DMSP/OLS and NPP/VIIRS satellite images to estimate the $CO_2$ emissions of 2735 counties (districts) in China from 1997–2017 [53], which provides good data support for this study. This study merges the $CO_2$ data of the corresponding city municipalities on the basis of the algorithm and constructs a dataset of $CO_2$ emissions in urban municipalities.

*3.2. Analysis Model*

3.2.1. Kernel Density Estimation Method

The kernel density estimation method is a non-parametric estimation method proposed by Rosenblatt and Parzen et al. [54,55]. It mainly describes the distribution pattern of random variables with the help of continuous density curves and avoids the subjectivity of parametric model estimation forms because it starts from the data itself and does not depend on the model setting. At the same time, it has the advantages of being less dependent on the length of the study and of allowing intuitive observation of the dynamic evolution of random variables and is widely used in the field of describing the unbalanced distribution of economic variables. In view of this, this paper uses the kernel density estimation method to explore the dynamic evolution characteristics of LCULUE, which is constructed as shown in the following paper [5].

Assuming that the random variables $X_1, X_2 \cdots X_n$ obey an independent identical distribution $F$, and that the density function of any point $X_i$ is $f(x)$, the empirical distribution function of the sample is then given as shown in Equation (6).

$$f(x) = \frac{1}{n}\{X_1, X_2, \ldots, X_n\}, \qquad (6)$$

The kernel density estimation formula is shown in Equation (7).

$$f(x) = \frac{[F_n(x+h_n) - F_n(x-h_n)]}{2h} = \int_{x-h_n}^{x+h_n} \frac{1}{h} K\left(\frac{t-x}{h_n}\right) dF_n(t) = \frac{1}{nh_n} \sum_{i=1}^{n} K\left(\frac{x-x_i}{h_n}\right), \tag{7}$$

In Equation (7), n is the number of samples, h is the smoothing parameter, indicating the bandwidth, And n and h satisfy: $\lim\limits_{n\to\infty} h(n) = 0$, $\lim\limits_{n\to\infty} nh(H) \to 0$. Here, $k(\cdot)$ is the form of the kernel density function, which can be classified into a triangular kernel, a quartic kernel, and a Gaussian kernel depending on the form of expression.

### 3.2.2. Tobit Model

To obtain information on how to improve the LCULUE, this study further explores the factors influencing the LCULUE. Considering that the values of LCULUE measured by the NDDF model range from 0 to 1, which has a significant truncation phenomenon, if the traditional linear model is used for estimation, the estimation results may be biased and the estimates inconsistent [56]. The Tobit model using the maximum likelihood method can be a good solution to this problem. Therefore, this study uses the Tobit model to analyze the factors influencing the LCULUE, and the specific expression is as follows (8) [56]:

$$y_i^* = \beta X_i u_i, u_i \sim N\left(0, \sigma_{u_i}^2\right), i = 1, 2, \dots, n$$
$$y_i = \begin{cases} y_i^*, y_i^* > 0 \\ 0, y_i^* \leq 0 \end{cases} \tag{8}$$

where $i$ represents $i$-th DMU, $y^*$ is the latent variable, $y$ is the dependent variable that is value LCULUE, $X$ is $K \times 1$ matrix which stands for independent variables, and $u$ is the stochastic error and submits to $N\left(0, \sigma_{u_i}^2\right)$. Based on existing studies, the following influencing factors are selected in this study.

(1) Land finance: The transaction of urban land is an important source of government revenue. Prompted by financial pressure, the government obtains funds by trading large amounts of urban land, resulting in rapid urban expansion and on the other hand invests the funds from land transactions into the construction of the national economy to promote the economic development of the region. The ratio of the transaction price of urban construction land concessions to the regional GDP is chosen to represent [57].

(2) Economic level: Economic level is closely related to land use efficiency, and regions with a high level of economic development usually have higher land use efficiency. The gross regional product per capita is chosen [1], and the natural logarithm is taken to mitigate the effects of dimensionality and heteroskedasticity.

(3) Population density: An increase in population density will promote the aggregation of resources, generating economies of scale and improving land use efficiency but may also increase the cost of congestion and environmental pressure, which may inhibit the improvement of urban land use efficiency. The number of people per unit area is chosen to represent [15], and in the same way, the natural logarithm is taken for it.

(4) Industrial structure: The secondary output value is characterized by high energy consumption and high emissions, and usually the higher the ratio of the secondary industry, the lower the urban land use efficiency. The ratio of the gross regional product of the secondary industry to the gross regional product is chosen.

(5) Level of transport facilities: The increase in the level of transport facilities increases the accessibility of space and also expands the urban area [38]. The area of actual urban roads per capita at the end of the year is chosen and treated as a natural logarithm.

### 3.3. Data Sources and Notes

The original data used for the indicators in this paper come from the *China City Statistical Yearbook, the China Statistical Yearbook, the China Land and Resources Statistical Yearbook* and science data. To avoid the impact of price changes, this study uses price

indices to transform the data into constant prices calculated in 2005 as the base period. Considering that there is no prefecture-level price index, this study uses the provincial price index for calculation. Missing values are supplemented by local statistical yearbooks or by interpolation. At the same time, to reduce the effect of extreme values in the regression analysis, the variables are subjected to a 1% tail reduction. In addition, the administrative boundaries of the municipal districts of prefecture-level cities in China change frequently, and the change of counties (county-level cities) to districts occurs frequently, so the statistical caliber and geographical scope of the data before and after the change of counties to districts are not consistent and not comparable. Therefore, the cities whose administrative areas of municipal districts changed between 2005 and 2016 were excluded from this paper, leaving the final 204 cities as the study population. The statistical information of each variable is shown in Table 2.

**Table 2.** Descriptive statistics.

| Variable | Obs | Mean | Std. Dev. | Min | Max |
| --- | --- | --- | --- | --- | --- |
| LCULUE | 2448 | 0.9 | 0.068 | 0.683 | 0.972 |
| Land finance | 2448 | 0.038 | 0.032 | 0.002 | 0.17 |
| Economic level | 2448 | 10.303 | 0.66 | 8.665 | 11.779 |
| Industrial structure | 2448 | 0.502 | 0.122 | 0.198 | 0.831 |
| Population density | 2448 | 7.94 | 0.831 | 5.642 | 9.37 |
| Level of transport facilities | 2448 | 2.194 | 0.573 | 0.762 | 3.791 |

## 4. Results

### 4.1. Trends in LCULUE

This study uses the maxDEA software to calculate the LCULUE of 204 cities from 2005 to 2016, and the averages and medians of the years are shown in Figure 1. It can be seen that from 2005 to 2016, the urban land use efficiency was fluctuating, with the mean value fluctuating above and below 0.9, without any significant deviation, reaching the peak in 2007, 2010 and 2012, respectively. The median trend is consistent with the mean and slightly higher than the mean, indicating that the distribution of urban land use efficiency is left-skewed. This result is very different from Liu et al.'s study, which showed that the LCULUE has been decreasing and is at a low level [15]. The high consistency between this study and Liu et al.'s study in terms of indicator selection and sample selection suggests that without separating out the effects of other input factors, the true picture of urban land use will not be reflected, and in fact, the reduction in all-factor urban land use efficiency is more likely to come from the concentration of labor and the increase in capital inputs.

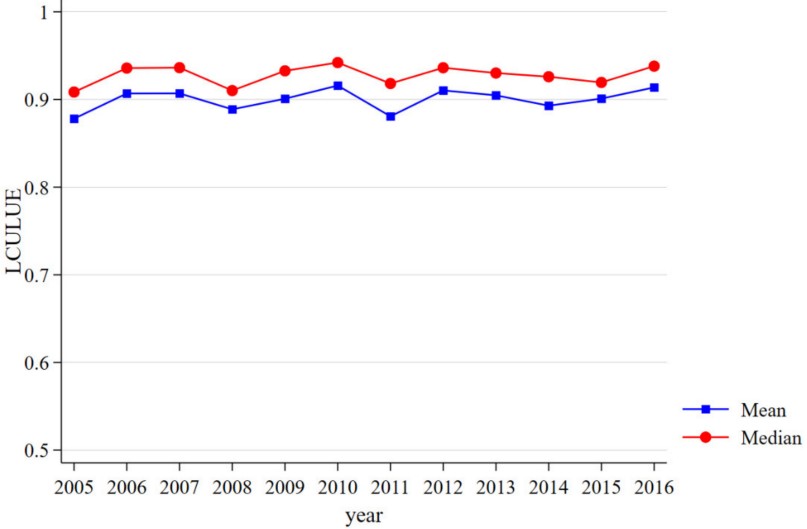

**Figure 1.** Mean and median of LCULUE over the years.

Next, we classified the cities into eastern, western and central regions, and made a graph of the historical averages of LCULUE for each region, as shown in Figure 2 [1]. It can be seen that in the early years, the eastern cities were in the lead in terms of LCULUE, which was gradually overtaken by the western cities, followed by a state of staggered development. The middle cities had lower efficiency values than the other two regions for a long time and then gradually caught up, with the gap between the efficiency values of each region gradually being narrowed.

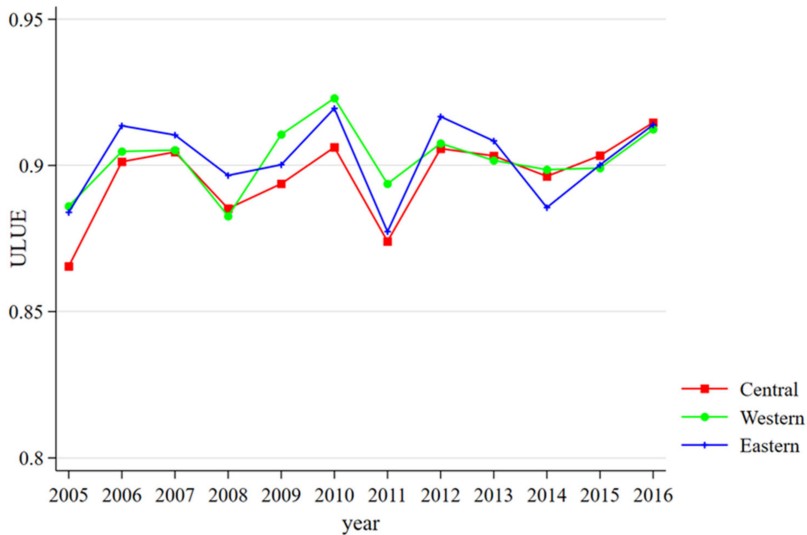

**Figure 2.** Average LCULUE by region over time.

### 4.2. Dynamic Evolutionary Patterns of LCULUE

Figure 3 shows the kernel density estimates for the country in 2005, 2009, 2013 and 2016. It can be seen from the graphs that the crest of the LCULUE is high in all years, and the density of the distribution with a high level is gradually increasing, showing that the crest position is close to 1 and shifts upwards and to the right over time, with the range of the right shift first decreasing and then increasing. The distribution curve gradually becomes steeper and narrower. This indicates that the proportion of low value areas in China's LCULUE has decreased, while the proportion of high value areas has increased, and the overall trend has maintained an upward trend. However, the speed of change has gone through a process of fast to slow to fast, and the gap between the efficiency values of each city is gradually narrowing, which is consistent with the above findings. Overall, the efficiency of low-carbon land use in China's cities has developed at different levels and at different rates over different time periods.

Figure 4 shows the kernel density estimates of LCULUE for each region. It can be seen that while the kernel density distribution map for the eastern region is highly similar to the national scale, the western and central regions have very different dynamic evolutionary characteristics. The western region showed a double peak in 2005, which disappeared with time and faded into the right wave, with the highest peak value appearing in 2009, and the peak experienced a process of moving up, then down and then up again. The magnitude of change gradually became smaller, indicating that the efficiency of the western region, although constantly going through a convergence–divergence process, still converged in the long run. The proportion of high-efficiency cities increased compared to the proportion of efficient cities is also increasing compared to the initial years. For the middle region, the highest peak occurred in 2016, and the peak has been rising. However, the magnitude of the rise has gradually decreased, and the density curve has shown a trend of "wide peak—sharp peak". This indicates that the efficiency values of the middle cities have been converging, but the convergence trend is slowing down, and the proportion of efficient cities is increasing.

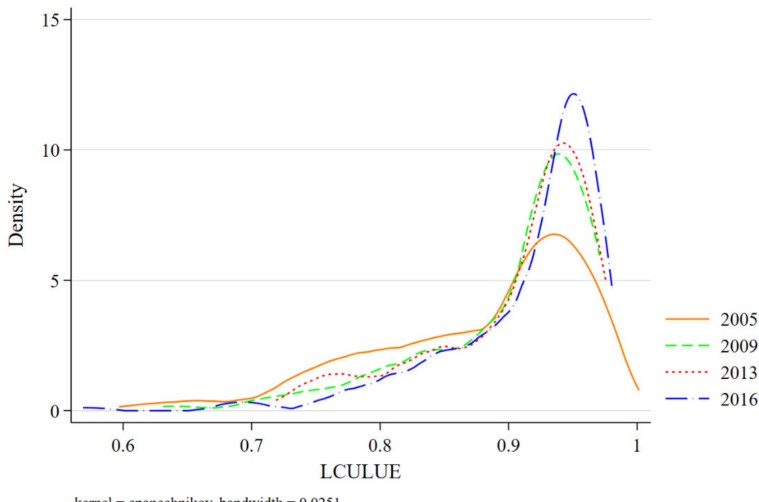

**Figure 3.** Kernel density estimates of LCULUE.

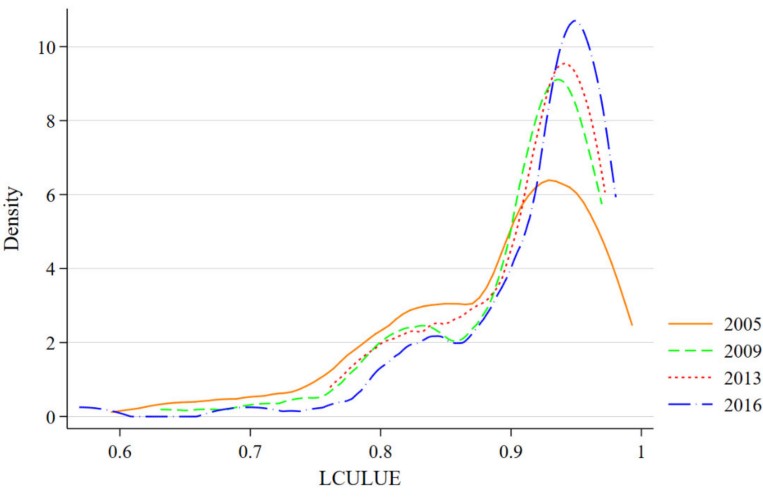

(**A**) Kernel density estimates of LCULUE (eastern region).

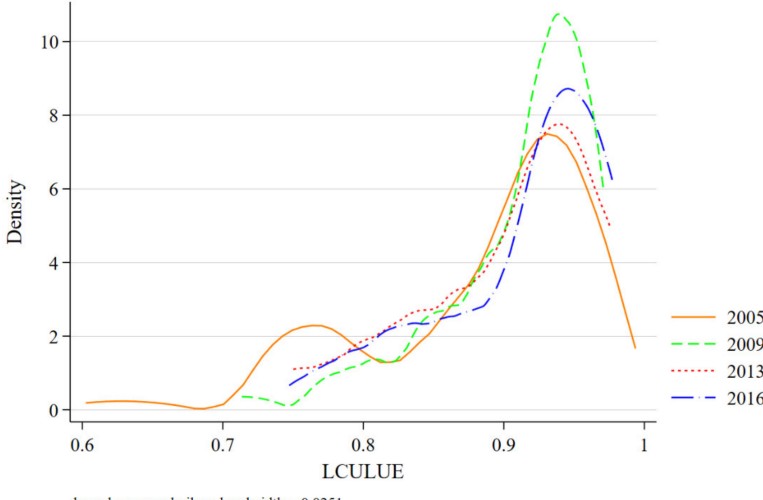

(**B**) Kernel density estimates of LCULUE (central region).

**Figure 4.** *Cont*.

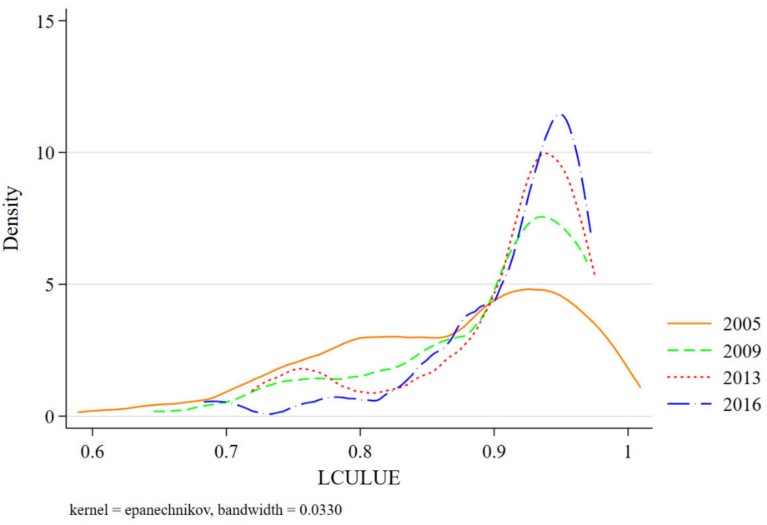

(**C**) Kernel density estimates of LCULUE (western region).

**Figure 4.** Kernel density estimates of LCULUE for each region.

*4.3. Analysis of Land Inputs and the Scope for Reducing Carbon Dioxide Emissions*

This study further calculates the target values of land inputs and $CO_2$ emissions and compares them with the actual values. Due to the limitation of the paper, only the proportion of land inputs and $CO_2$ emissions that can be reduced is shown in Table 3. The results found that there is a lot of room for reduction in land input and carbon dioxide emissions without affecting economic output. In 2016, for example, if all cities in the sample achieved efficient production, the land input of all cities in the sample could be reduced by 2435.2 km$^2$, which is equivalent to the combined built-up area of Beijing and Tianjin, and $CO_2$ emissions could be reduced by 113.78 Mt, accounting for 10.38% and 5.31% of the total land input and $CO_2$ emissions in the sample, respectively.

**Table 3.** The proportion of land and $CO_2$ emissions that can be reduced.

| Year | All | | Eastern | | Western | | Middle | |
| --- | --- | --- | --- | --- | --- | --- | --- | --- |
| | Land | $CO_2$ | Land | $CO_2$ | Land | $CO_2$ | Land | $CO_2$ |
| 2005 | 8.09% | 5.34% | 7.85% | 3.93% | 8.31% | 2.89% | 8.32% | 9.38% |
| 2006 | 9.18% | 4.41% | 8.16% | 3.91% | 11.92% | 3.4% | 9.12% | 5.95% |
| 2007 | 7.99% | 5.29% | 7.97% | 5.07% | 10.43% | 5.6% | 6.58% | 5.44% |
| 2008 | 9.96% | 4.87% | 9.09% | 5.08% | 11.19% | 3.86% | 10.57% | 5.3% |
| 2009 | 8.71% | 4.83% | 9.03% | 4.41% | 9.15% | 4.4% | 7.98% | 5.83% |
| 2010 | 6.53% | 5.65% | 6.43% | 5.12% | 6.45% | 3.83% | 6.72% | 7.88% |
| 2011 | 9.42% | 4.77% | 9.6% | 4.81% | 7.2% | 4.42% | 10.43% | 5% |
| 2012 | 6.68% | 5.47% | 4.96% | 5.62% | 8.32% | 5.4% | 8.18% | 5.3% |
| 2013 | 9.19% | 4.82% | 7.25% | 5.03% | 13.36% | 3.64% | 9.38% | 5.48% |
| 2014 | 9.31% | 5.24% | 10.43% | 5.72% | 6.93% | 5.06% | 9.04% | 4.66% |
| 2015 | 9.97% | 5.05% | 9.9% | 4.57% | 10.72% | 5.69% | 9.59% | 5.26% |
| 2016 | 10.38% | 5.31% | 10.92% | 6.03% | 11.67% | 4.47% | 8.72% | 4.89% |

In terms of regions, the reduction in land input is most significant in the west, followed by the middle and is lowest in the east. This is closely related to China's land policy. For the sake of balanced development, the central government has implemented a land supply policy that favors the central and western regions and adopts a stricter management of construction land targets in the eastern regions, thus causing an excessive expansion of urban land in the central and western regions, resulting in a significant deterioration of

land use efficiency in the central and western regions [58]. In terms of the space available for $CO_2$ emission reduction, the central region has the highest, the eastern region the second highest and the western region the smallest. This is probably due to the transfer of highly polluting and high-emission industries from the east to the mainland, with the central region taking over a large number of high-emission industrial enterprises, resulting in greater scope for reducing $CO_2$ emissions.

### 4.4. Analysis of Factors Influencing LCULUE

This study uses stata 15.0 to run regressions, and the results are shown in Table 4. At the nationwide level, the economic level, land finance, population density and transport facility level all pass the significance level test, while the industrial structure does not pass. Land finance, economic level and population density have a positive effect on the LCULUE, while the level of transport facilities has a negative effect. In terms of economic significance, for every 0.01 increase in land finance, LCULUE increases by 0.00125; for every 1% increase in economic level and population density, LCULUE increases by 0.0111 and 0.0046, respectively, while for every 1% increase in transport facilities, LCULUE decreases by 0.0088. Land finance has been criticized as an inefficient use of urban land. In fact, the contribution of the government's model of generating money from land and investing the revenue in economic development to urban land use and urban development should not be ignored, but its negative impact on urban land use efficiency may be more due to the over-investment caused by duplication of construction. The positive effects of land finance can be seen when we control fixed asset investment and labor. The positive effects of economic development and population agglomeration on LCULUE suggest that economic development is still an important means of improving LCULUE, while there is still potential for the agglomeration of population and resources to increase, with the positive effects outweighing the negative effects. The negative impact of the level of transport facilities is mainly attributed to the increase in road space per capita due to the expansion of the urban extension, which is less utilized, while the road space in the urban center, which has a higher utilization rate, has not been effectively improved. In addition, the increase in transport facilities also means an increase in transport modes, which promotes carbon emissions and can also reduce the LCULUE.

**Table 4.** Regression results of factors influencing LCULUE.

| | All | Eastern | Middle | Western |
|---|---|---|---|---|
| **Variable** | **LCULUE** | **LCULUE** | **LCULUE** | **LCULUE** |
| Land finance | 0.125 ** | 0.136 * | 0.130 | 0.134 |
| | (0.0503) | (0.0713) | (0.0898) | (0.129) |
| Economic level | 0.0111 *** | 0.000194 | 0.0175 ** | 0.0163 ** |
| | (0.00391) | (0.00644) | (0.00703) | (0.00724) |
| Industrial structure | −0.00774 | 0.0132 | −0.0385 | −0.0101 |
| | (0.0168) | (0.0279) | (0.0318) | (0.0281) |
| Population density | 0.0046 ** | 0.00491 | 0.00725 * | 0.00379 |
| | (0.00203) | (0.00328) | (0.00433) | (0.00318) |
| Level of transport facilities | −0.0088 ** | −0.00357 | −0.00863 | −0.0133 * |
| | (0.00429) | (0.00705) | (0.00864) | (0.00704) |
| Constant | 0.768 *** | 0.859 *** | 0.691 *** | 0.735 *** |
| | (0.0357) | (0.0608) | (0.0653) | (0.0621) |
| Observations | 2448 | 912 | 852 | 684 |
| Number of Cities | 204 | 76 | 71 | 57 |

Note. Standard errors in parentheses. *** $p < 0.01$, ** $p < 0.05$, * $p < 0.1$.

In terms of individual regions, there is significant regional heterogeneity in the impact effect. In the east, only land finance passes the significance level test and is positively

influenced, while in the middle and west, land finance is not significant. This indicates that land supply policies are biased towards the middle region, where they do not have the expected effect, but instead contribute to the inefficient use of urban land in the region, which is consistent with the previous results. In contrast to the middle region, the eastern region needs more land, and land finance can play its positive role better. For the middle region, economic development and population density are significantly positive, indicating that the middle region still needs to focus on economic development and actively promote the agglomeration of population and resources to promote the efficient use of urban land. For the western region, economic development plays a positive role, while the level of transport facilities plays a negative role, indicating that while the western region is vigorously developing its economy, it also needs to consider the need to optimize the spatial distribution of land in the built-up areas of the city and avoid the inefficient use of transport infrastructure.

## 5. Conclusions and Policy Recommendations

Urban land use is the main cause of the rapid growth of $CO_2$ emissions. Therefore, the achievement of low-carbon urban land use in China is not only conducive to the country's sustainable development, but also helps China to achieve international commitments on carbon emission reduction. In this study, an evaluation system for the LCULUE was constructed using relevant data from 204 prefecture-level cities in China from 2005 to 2016. The non-radial directional distance function is then used to measure the LCULUE of each city, and based on that, its dynamic evolution pattern and influencing factors are analyzed, and several conclusions are drawn as follows:

(1) The LCULUE in China is generally fluctuating above and below 0.9, and the gap between the LCULUE of various cities is narrowing and tending to converge.

(2) There is much potential to reduce land input and carbon emissions in the Chinese cities. In 2016, land input and carbon emissions in the sample could be reduced by 10.38% and 5.31% respectively, with greater potential for compression in the mid-west.

(3) At the nationwide level, land finance, economic level and population density have a positive effect on LCULUE, while traffic levels have a negative effect, and these effects show significant regional heterogeneity.

Based on the above conclusions, this study puts forward the following policy recommendations. First, the government should improve the system of assessment indicators for officials as soon as possible, incorporating indicators such as factor inputs and pollutant outputs based on economic performance, to prevent wastage of resources such as land and pollution of the environment through increased factor inputs by officials in pursuit of economic benefits. Secondly, the government should reform the current system of allocating construction land indicators [2] and hand them over to the market for allocation, for example, by building a trading platform for construction land indicators, where the central government is the initial allocator of construction land indicators, so that through market forces construction land indicators can be transferred from inefficient cities to high-efficiency cities, and the proceeds from the sale of construction land indicators can be used for local construction and economic development. Thirdly, as China is a vast country with great regional differences, local governments should tailor their policies to local conditions and not blindly copy the practices of other regions, especially in the mid-west, where the government should give up the land for wealth approach and promote the agglomeration of resources.

In addition, there are some limitations in this study, which can be fixed in future research. Firstly, this study mainly focuses on carbon emissions as a non-desired output and economic output in the evaluation index system. Research in the future may expand the types of outputs to reflect the urban land use situation more comprehensively. Secondly, this study does not systematically explore the dynamic evolution of LCULUE in green cities and the underlying causes of regional differences. Thirdly, urban land use efficiency is driven by a combination of factors, and the influencing factors explored in this study may

only be part of the factors, and it is not known if there are other factors that influence it and their mechanisms of action. At the same time, this study focuses on the independent effects of the influencing factors on LCULUE, and the interaction mechanisms involving two or more influencing factors are less explored, which will be the focus of our next phase of research. Considering the limitations of this study, the policy recommendations presented in this study are not yet mature.

**Author Contributions:** Conceptualization, C.M.; Data curation, H.C.; Formal analysis, H.C.; Funding acquisition, Q.C.; Methodology, H.C. and C.M.; Software, C.M.; Supervision, Q.C.; Writing—original draft, H.C. and C.M.; Writing—review & editing, H.C. and Q.C. All authors have read and agreed to the published version of the manuscript.

**Funding:** This research was funded by [the National Natural Science Foundation of China] grant number [No. 71072066], [Sichuan University] grant number [No. 2019hhf-14], [the Sichuan Soft Science Research Project] grant number [No. 2019JDR0345], [the Social Science Planning Project of Sichuan Province] grant number [SC19TJ026].

**Institutional Review Board Statement:** Not applicable.

**Informed Consent Statement:** Not applicable.

**Data Availability Statement:** Some of the data is sourced from paid databases. If you require data on the manuscript, please contact the first author. The original data of the manuscript will be provided to you after obtaining the authorization of the copyright holder.

**Conflicts of Interest:** The authors declare no conflict of interest.

## Notes

1. The regional classification is based on the "East-West-Central and Northeast Regional Classification Methodology" of the National Bureau of Statistics of China, in which the Northeast and the East are unified as the Eastern Region in this study, the specific classification method http://www.stats.gov.cn/ztjc/zthd/sjtjr/dejtjkfr/tjkp/201106/t20110613_71947.htm (accessed on 19 May 2022).

2. Local governments are only eligible to convert agricultural land into construction land once they have been given construction land targets. The existing construction land targets are decentralized from the central government to local governments through the scale of construction land in the land use plan, and the number of construction land targets determines the scale of construction land.

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
