# Peer review of "Measurement and Influencing Factors of Low Carbon Urban Land Use Efficiency—Based on Non-Radial Directional Distance Function"

_land, doi:10.3390/land11071052_

Round 1

Reviewer 1 Report

The authors construct an evaluation system for the Low Carbon Urban Land Use Efficiency (LCULUE) by taking carbon emissions as a non-desired output, measure the LCULUE using a non-radial directional distance function based on the statistical data of 204 cities in China from 2005 to 2016, and explore its dynamic evolution pattern and influencing factors using the kernel density estimation method and Tobit model. The contributions of the present paper are clearly highlighted. It is an interesting and well written paper and in my opinion could be published. 

Some minor suggestions are listed below:

1. "At present, a great number of studies have been conducted on urban land use efficiency in academia.": Please provide some examples as references.

2. "From the connotation of urban land use efficiency to the measurement method, there are still many theories, but it is hard to say which one is better.": Please refer to some of these theories.

3. Please provide explanation for all symbols in equations. There are some missing.

Author Response

Dear Reviewer,

Hope everything goes well with you!

Thank you very much for giving us the opportunity to have our manuscript subject to peer review and to revise our manuscript according to the comments of you. We have modified our paper accordingly, as is shown in the paper with red text.

We answer your concerns as follows:

Q1:"At present, a great number of studies have been conducted on urban land use efficiency in academia.": Please provide some examples as references.

Reply: This sentence serves as a summary and guide to what follows below, and in fact the section that follows is an example of the main findings to date.

Q2: "From the connotation of urban land use efficiency to the measurement method, there are still many theories, but it is hard to say which one is better.": Please refer to some of these theories.

Reply: This sentence complements the first one by emphasizing the variety of current research methods. The second half focuses on the various methods and their differences.

Q3: Please provide an explanation for all symbols in equations. There are some missing

Reply: Additions have been made, see line 159-160, line 190-192, line 223-224 and line295-296.

Based on comments from other reviewers, we have made the following changes to this paper.

  1. Improved the data on arable land per capita in China and the world, see line 40-43.
  2. Some repetitive sentences have been removed, see line 97 and line 135-137.
  3. We added the relationship between inputs and outputs in line 179-189.
  4. The names of the variables in tables 2 and 4 have been fixed.
  5. The title of Table 3 has been fixed.
  6. In the form of a footnote we have added an explanation of the construction land indicator, see footnote on page 15.
  7. Emphasized at the end of the article that the policy recommendations are only preliminary, see, line 514-515.
  8. Added missing information from the literature [3], see, line 530-532.

Thank you again for your help. We are very anxious for the manuscript to be published in this high-quality journal.

If you have any questions or requirements, please feel free to contact me.

Kind regards,

Authors

Reviewer 2 Report

The paper is appropriate for the Land journal and proposes a methodology to measure Low Carbon Urban Land Use Efficiency index in Chinese cities. The analysis of the index spatial and temporal (2005-2016) dynamic evolution and its influencing factors in a sample of 204 Chinese cities, bring the authors to suggest some policy recommendations to promote an efficient urban land use, at the national level, and achieve low carbon economic development.

The manuscript is generally clear and relevant for the field, and discusses issues that I find significant and not yet resolved, i.e. how to balance territorial economic development, which also depends on urban land uses, and the environmental protection and climate change mitigation. Reasoning about the efficiency of urban land use is certainly a research avenue, but I agree with the authors that a multi-factor assessment is needed (exploring more desired/undesired inputs and outputs), delving into the environmental impacts of urban land use, which are much more complex than emissions alone, as the authors rightly point out in their conclusions. (Possible priority issues to be addressed could be climate change adaptation, the undesired loss of ecosystem services and social issues).

The article is well structured and the whole experimental process is scientifically sound and well designed. The references are mostly recent and relevant to the study and figures and tables are appropriate to explain results.

The methodological approach is clear and appropriate to test the initial questions, but I think parameters in equation 2 need further clarification: for example, is the meaning of i and t, explained after equation 4 (p. 5 lines 210-211), the same as in equation 2? If yes, the explanation needs to be brought forward. 

In the conclusion, I really appreciated the fact that the authors highlighted the limitations of the study and the opportunities for future research. Perhaps, since many aspects related to urban land use efficiency, especially input/output indicators, have not yet been evaluated, as the authors themselves state, the policy recommendations (p.14 from line 475 to line 490) could be presented in a less conclusive manner. I suggest presenting them as initial suggestions deduced from the study, insisting instead on the need for reliable data, postponing further evaluation and recommendations to later stages of the research.

In discussing the study limitations/opportunities, perhaps, the authors may consider also the following additional issues, e.g.:

-         Concerning the sample of 204 cities: which criteria and which relevant cities, if any, were excluded from the study and whether it would be worthwhile to test the methodology again on a different sample.

-         About the time frame investigated (2005-2016): is it possible to work on more recent data? Are these data missing or simply unavailable?

-         Regarding the scaling up: is the methodology applicable to other countries? If so, under what conditions could it be applied?

Finally, I suggest few specific revisions concerning unclear sentences and inaccuracies within the text and references:

-         In Table 3 – caption is unclear and should be revised.

-         Reference [3] lacks information on authors, time and source.

-         p.1 line 41: I would really appreciate if authors reported in the text the figures of per capita amount of arable land compared to the world average.

-         p. 2 line 96 and p. 3 line 135: repeated sentences  

-         p. 3 line 98: missing capital letter after period

-         p.14 line 442: missing space after Table 3

-         p.14 line 469: sentence unclear (‘in China’ or ‘in the Chinese cities’?)

Reviewer 3 Report

A) In general, the paper must describe in depth the specific connection between each input and the correspondent output.

B) Lines 235 and Table 1 – Labour: it could be strong differences between secondary and tertiary industry about the effects on CO2.

C) In Table 2 and in Table 4 – variables must be write in a first column as a full word.

D) It is not specified how capital can affect the CO2. Explain

E) It is not clear what are land indicators (e.g., line 481; 483). Explain

F) References are quite only local (China). Tha parers laks of interenational references.

Author Response

Dear Reviewer,

Hope everything goes well with you!

Thank you very much for giving us the opportunity to have our manuscript subject to peer review and to revise our manuscript according to the comments of you. We have modified our paper accordingly, as is shown in the paper with red text.

We answer your concerns as follows:

Q1: In general, the paper must describe in depth the specific connection between each input and the correspondent output.

Reply: In line 179-189 we add the relationship between inputs and outputs

Q2: Lines 235 and Table 1 – Labour: it could be strong differences between secondary and tertiary industry about the effects on CO2.

Reply: In this study, we measure the efficiency level of total inputs and therefore do not break them down. In fact, there are similar studies that refine the classification of land (labour) by looking at the efficiency of land use in a particular area of the city, for example, the efficiency of industrial land use. However, more studies have focused on the level of efficiency of total inputs, which is the original intention of this study, and it would be inconsistent with the theme of this study to refine the classification. Future research by this group will explore the gaps in the efficiency of different types of land (labour) and the factors influencing them.

Q3: In Table 2 and in Table 4 – variables must be written in a first column as a full word.

Reply: fixed, see line 337 and line 468.

Q4: It is not specified how capital can affect the CO2. Explain

Reply: This study focuses on the desired output (economic) and the undesired output (CO2) generated in the process of urban land use. In fact, all economic activities on urban land as a spatial carrier for urban development generate CO2. In this study, capital is only one of the input factors and its use is part of urban land use, which means that capital use cannot exist separately from land use. Therefore, this paper focuses on the relationship between urban land use and CO2 emissions but does not elaborate on the impact of capital on CO2 emissions.

Q5: It is not clear what are land indicators (e.g., line 481; 483). Explain

Reply:We have added an explanation of the construction land indicators in the form of a footnote, see footnote on page 15.

Q6: References are quite only local (China). Tha parers laks of interenational references.

Reply:Almost all of the references in this study are from international journals. As this study focuses on urban land issues in China, such issues have received more attention from Chinese scholars and a great deal of work has been done, which has provided us with many ideas and content to draw on, so this study draws mostly on Chinese scholars, while we also draw on the experience of other international scholars.

Based on comments from other reviewers, we have made the following changes to this study.

  1. added data on arable land per capita in China and the world, see line 40-43.
  2. some repetitive statements have been removed, see line 97 and line 135-137.
  3. Changed the title of Table 3.
  4. added explanation of characters in the public notice, see line 159-160, line 190-192, line 223-224 and line295-296.
  5. emphasized at the end of the article that the policy recommendations are only premature, see, line 514-515.
  6. added missing information from literature [3], see, line 530-532.

Thank you again for your help. We are very anxious for the manuscript to be published in this high-quality journal.

If you have any questions or requirements, please feel free to contact me.

Kind regards,

Authors